# Cytosolic Phospholipase A2 Is Required for Fexofenadine’s Therapeutic Effects against Inflammatory Bowel Disease in Mice

**DOI:** 10.3390/ijms222011155

**Published:** 2021-10-15

**Authors:** Xiangli Zhao, Ronghan Liu, Yuehong Chen, Aubryanna Hettinghouse, Chuanju Liu

**Affiliations:** 1Department of Orthopaedic Surgery, New York University Medical Center, New York, NY 10003, USA; Xiangli.Zhao@nyulangone.org (X.Z.); qq228660329@gmail.com (R.L.); yuehongchen@scu.edu.cn (Y.C.); Aubryanna.Hettinghouse@nyulangone.org (A.H.); 2Department of Cell Biology, New York University School of Medicine, New York, NY 10016, USA

**Keywords:** inflammatory bowel disease, fexofenadine, TNFα, histamine H1 receptor, cPLA2

## Abstract

Inflammatory Bowel Disease (IBD) is an autoimmune condition with complicated pathology and diverse clinical signs. TNFα is believed to play a crucial role in the pathogenesis of IBD. We recently identified fexofenadine, a well-known antagonist of histamine H1 receptor, as a novel inhibitor of TNFα signaling. Additionally, cytosolic phospholipase A2 (cPLA2) was isolated as a binding target of fexofenadine, and fexofenadine-mediated anti-TNF activity relied on cPLA2 in vitro. The objective of this study is to determine whether fexofenadine is therapeutic against chemically-induced murine IBD model and whether cPLA2 and/or histamine H1 receptor is important for fexofenadine’s anti-inflammatory activity in vivo by leveraging various genetically modified mice and chemically induced murine IBD models. Both dextran sulfate sodium- and 2, 4, 6-trinitrobenzene sulfonic acid-induced murine IBD models revealed that orally delivered fexofenadine was therapeutic against IBD, evidenced by mitigated clinical symptoms, decreased secretions of the proinflammatory cytokine IL-6 and IL-1β, lowered intestinal inflammation, and reduced p-p65 and p-IĸBα. Intriguingly, Fexofenadine-mediated protective effects against IBD were lost in cPLA2 deficient mice but not in histamine H1 receptor-deficient mice. Collectively, these findings demonstrate the therapeutic effects of over-the-counter drug Fexofenadine in treating DSS-induced IBD murine and provide first in vivo evidence showing that cPLA2 is required for fexofenadine’s therapeutic effects in murine IBD model and probably other inflammatory and autoimmune diseases as well.

## 1. Introduction

Autoimmune diseases include a wide range of conditions characterized by loss of immune self-tolerance and target organ attack [1]. Inflammatory Bowel Disease (IBD), including ulcerative colitis and Crohn’s disease, is one of the most common autoimmune diseases [2]. IBD pathogenesis involves a complex interplay between the intestinal epithelium, exaggerated immune response, and intestinal microorganisms [3]. Pathology presents diverse clinical signs, which are alterations in cytokine expression and immune cell infiltration [4,5]. Among the proinflammatory cytokines involved, tumor necrosis factor-alpha (TNFɑ) has received great attention due to its position at the apex of the proinflammatory cytokine cascade [3,4].

TNFα is involved in the onset and progression of many inflammatory diseases, and elevated expression of TNFα is believed to play an important role in the pathogenesis of IBD [5]. Treatments of IBD patients with TNFα chimeric monoclonal antibodies caused an extensive investigation of TNFα’s role in IBD [6,7,8]. Anti-TNFα therapy has been approved to use in Crohn’s disease since 1998, including monoclonal antibodies or fragments thereof directed against TNF molecules, such as Infliximab (IFX), Adalimumab (ADA), Certolizumab pegol (Cimzia) [9,10]. Although anti-TNFα agents have proven efficacious treatments in immune-mediated inflammatory conditions, not all patients respond to treatment [7,11]. In addition, inhibition of TNF activity may also contribute to carcinogenesis [12,13,14]. Thus, identifying and characterizing novel, safer and cost-effective antagonists of TNFɑ is of great importance from both pathophysiological and therapeutic standpoints. Because drug development is time-consuming and extremely expensive, we previously screened an FDA-approved drug library using TNFα/NF-κB reporter constructs and mice [15]. Our screens led to the identification of Terfenadine (TFD) and its active metabolite Fexofenadine (FFD) as novel inhibitors of TNFα signaling [15].

TFD, known as histamine receptor antagonists developed to treat allergic conditions [16], is the first-generation antihistamine drug and has been associated with potentially life-threatening arrhythmias and has been pulled from the market. In contrast, FFD, the predominant active metabolite of terfenadine and a second-generation selective non-sedating histamine antagonist, lacks cardiotoxicity and drug interaction potential [17]. FFD is safer (non-prescription medicine), more convenient (taken orally), cost-effective, and has been used widely to treat various allergic diseases [18,19]. In addition, potential therapeutic effects of FFD for the treatment of inflammatory bowel disease have been explored in murine colitis models induced by DSS [19,20], but the molecular mechanisms underlying remains unelucidated.

Combined use of drug affinity responsive target stability assay [21] and proteomics, we recently reported cytosolic phospholipases A2 (cPLA2) as a new target of FFD [15]. Moreover, FFD’s anti-TNF activity depends on cPLA2 in vitro [15]. Herein we found that TFD and FFD are therapeutic against the DSS-induced murine IBD model, illustrated through reduced clinical symptoms of IBD, decreased secretion of IL-6 and IL-1β, decreased intestinal inflammation, and reduced p-p65 and p-IĸB-α expression levels in both dextran sulfate sodium (DSS)- and 2, 4, 6-trinitrobenzene sulfonic acid (TNBS)-induced experimental IBD mice. Additionally, implementation of our IBD models in histamine H1-receptor (H1R) knockout (KO) and cPLA2 KO mice, we clearly demonstrated that the therapeutic effect of FFD on IBD was predominately dependent on cPLA2, but not on its well-known target H1R. 

## 2. Results

### 2.1. Therapeutic Effects of Fexofenadine and Terfenadine in DSS-Induced Inflammatory Bowel Diseases

The murine DSS-induced IBD model, considered highly comparable to human colitis, was established via ad libitum access to drinking water containing 3% DSS. FFD or TFD was orally delivered beginning 3 days before model induction and continuing daily for a total of 10 days. 5-aminosalicylic acid (5-ASA, 50 mg/kg), which has been used for treating IBD for over 30 years [22], was employed as a positive control [23]. The onset and progression of IBD were assessed by four readouts: bleeding, body weight, stool, and colon length. We found that both FFD and TFD could prevent bleeding (Figure 1a,e), bodyweight loss (Figure 1b,f), abnormal stool (Figure 1c,g), and colon shortening (Figure 1d,h) in DSS-induced IBD mice. Interestingly, the efficacy of TFD for preventing clinical signs of IBD increased in a dose-dependent manner; however, FFD had most efficacy at a specific mid-range dose (2 mg/kg) (Figure 1).

IBD is characterized by chronic inflammation involving numerous proinflammatory cytokines such as TNFα, interleukin (IL)-6, IL-12, IL-23, IL-17, and IL-1β. Among these inflammatory molecules, TNFα is the main cytokine produced by innate immune cells, such as macrophages, monocytes, and differentiated T cells [24,25]. TNFα exerts its proinflammatory effects through increased production of IL-6 and IL-1β [26]. Since our recent screening found that FFD was an inhibitor of TNFα/NF-ĸB signaling, we next tested the secretion level of IL-1β and IL-6 in the serum of IBD model mice following TFD or FFD treatment. As shown in Figure 2, the serum levels of IL-1β and IL-6 were both reduced in FFD and TFD treatment groups compared to the vehicle group (Figure 2a–d). Notably, IL-1β secretion was lower in FFD- and TFD-treated groups compared with the positive control 5-ASA-treated group. FFD and TFD dose-dependently reduced IL-1β level; interestingly, secretion of IL-6 was irresponsive to low-dose FFD (0.4 mg/kg), but high-dose (2 mg/kg and 10 mg/kg) FFD was associated with a dramatic decrease in IL-6 levels. Histological analysis indicated the significant protective effects of FFD and TFD in DSS-induced IBD mice (Figure 2e–g).

### 2.2. Protective Effect of Fexofenadine and Terfenadine in TNBS-Induced Inflammatory Bowel Diseases

To ascertain whether the above observations were specific to the DSS-induced colitis model, another experimental colitis model which resembles Crohn’s disease was established through intrarectal administration of TNBS [27]. FFD or TFD was orally delivered 3 days before model induction, and body weight was recorded daily. As observed in the DSS-induced IBD model, FFD and TFD prevented both bodyweight loss and colon shortening (Figure 3a–e). Histological analysis indicated the significant protective effects of FFD and TFD in TNBS-induced IBD mice (Figure 3c–h). These results further confirmed the protective effect of TFD and FFD in chemically induced IBD mice models.

### 2.3. FFD and TFD Inhibited the Phosphorylation of NF-κB p65 and IκBα in DSS- and TNBS-Induced IBD Model

The nuclear transcription factor kappa B (NF-κB) is a master regulator of the inflammatory response through transcriptional control of genes that drive inflammation, cell development, proliferation, cycle, and death [28]. Inducible NF-κB activation depends on the convergence of diverse signaling pathways on the IκB kinase (IKK) complex [29]. The phosphorylation of IκB is essential for the signal transduction to NF-κB, which leads to the phosphorylation of NF-κB, and then regulates the transcription of target genes [29]. Multiple studies have suggested that in IBD patients, the activation of NF-κB promotes the production and secretion of proinflammatory cytokines TNF-ɑ, IL-6, and IL-1β [30,31]. Since we found the secretion level of IL-6 and IL-1β were dramatically decreased after FFD or TFD treatment in our IBD mice models, we assessed the level of phospho-p65 and phospho-IκBɑ via immunohistochemistry staining of FFD- or TFD-treated IBD model mice. Staining results show that FFD and TFD inhibited the phosphorylation of NF-κB p65 and its upstream mediator IκBα in both DSS- and TNBS-induced IBD mice (Figure 4).

### 2.4. H1R Is Not Indispensable for the Therapeutic Effects of FFD against IBD

It is well-known that FFD ameliorates allergic disease by targeting H1R [32]. Our recent study unexpectedly identified cPLA2 as a novel target of FFD [15,33]. Accordingly, we sought to examine the mechanism underlying protective effects of FFD observed in IBD and, more specifically, the relative importance of H1R and cPLA2 as targets of FFD in IBD. To this end, H1R KO mice, cPLA2 KO, and Wild-Type (WT) mice were subjected to DSS-induced IBD. Considering the potential proarrhythmic risk of TFD, FFD treatment alone was employed moving forward. H1R KO mice, cPLA2 KO, and WT mice were randomly assigned to one of four treatment groups: 5-ASA (50mg/kg) group served as a positive control, vehicle group was treated with FFD-vehicle, and FFD-treated mice were subjected to the best-working dose (2 mg/kg) prior to induction of the DSS-model; no induction group mice were maintained experimentally naïve. As shown in Figure 5, the bleeding and stool scores were significantly higher in the vehicle group than the no induction group in both WT and H1R KO mice. Moreover, the body weight was lost obviously in the vehicle group, and the colon length was significantly shortened in the vehicle group. These observations suggested that the IBD model was induced successfully. In WT mice, both treatments of FFD or 5-ASA effectively decreased IBD onset and progression (Figure 5a–c,g,h). Similar results were also found in H1R KO mice, i.e., the treatment with FFD could decrease IBD onset and progression effectively, including decreased the bleeding and stool score, less the body-weight loss, and rescued the shortened-colon (Figure 5d–g,i). In addition, we also investigated the effect of H1R deficiency on the development of IBD and found that H1R deficiency did not induce any difference in clinical symptoms between H1R KO mice and WT mice (Appendix A). These results indicated that FFD’s therapeutic effects in IBD do not depend on its well-known target H1R.

We next measured the secretion levels of inflammatory cytokines IL-6 and IL-1β in DSS-induced WT mice or H1R KO mice after FFD treatment. The secretion levels of IL-6 and IL-1β decreased after FFD or 5-ASA treatment compared with the vehicle group in WT mice (Figure 6a,c). Interestingly, in H1R deficient mice, although FFD could decrease IL-6 and IL-1β secretion levels, FFD was less effective in this regard relative to 5-ASA (Figure 6b,d). Analysis of histological scores clearly showed that FFD could prevent inflammation in WT mice and H1R KO mice (Figure 6e–h). These results suggested that FFD’s inhibition of proinflammatory cytokine secretions and protective effect against tissue damage in DSS-induced IBD murine may not be H1R-dependent.

### 2.5. cPLA2 Is Essential for the Protective Effects of FFD in IBD

cPLA2 was found as a novel target of FFD, and FFD lost its anti-TNF action in cPLA2 knockdown cells [15]. Since FFD’s effects did not, or at most partially, depend on H1R in the IBD model, we hypothesized that FFD effects in IBD depended primarily on cPLA2. We established the IBD model in cPLA2 KO mice and wild-type controls with or without FFD or 5-ASA administration to test this hypothesis. Both FFD and 5-ASA treatment reduced the bleeding and stool scores, bodyweight loss, and colon shortening in WT mice (Figure 7a–c,g,h). However, in cPLA2 deficient mice, the protective effects of FFD were largely impaired, indicated by failure to rescue clinical symptom scores and shortened clone length (Figure 7d–g,i). Interestingly, 5-ASA treatment was still effective in reducing clinical signs of IBD in cPLA2 KO mice (Figure 7d–g,i). Additionally, secretion levels of IL-6 and IL-1β were reduced after FFD and 5-ASA treatment in WT mice (Figure 8a,c). However, in cPLA2 deficient mice, FFD could not reduce IL-6 secretion level, and IL-1β secretion level was surprisingly increased, with FFD treatment during the IBD induction (Figure 8b,d). Histological analysis showed that FFD could prevent inflammation in WT mice, but this effect was completely lost in cPLA2 KO mice (Figure 8e–h). The 5-ASA treatment largely rescued the histological level in WT mice. Interestingly, in cPLA2 mice, 5-ASA treatment slightly rescued the histological level (Figure 8f,h). Collectively, these results indicate that the protective effect of FFD in IBD is largely cPLA2-dependent.

## 3. Discussion

TNFα is recognized as a marker of IBD, and its inhibitors are utilized for clinical treatment. To identify novel, safer, and more cost-effective antagonists of TNFɑ, our lab recently completed screening of a robust FDA-approved drug library and identified TFD and its active metabolite FFD as inhibitors of TNFα signaling [15]. Although TFD and FFD have traditionally been used in treating allergic diseases as histamine receptor antagonists [32], we present data supporting the potential utility of FFD for the treatment of inflammatory bowel disease [20].

TFD and FFD both show a preventative effect in DSS- and TNBS- induced IBD models testified through decreased clinical symptoms, including conserved colon length and reduced bleeding, body weight, and stool scores (Figure 1). It has been demonstrated that TNFα is an early potent proinflammatory cytokine in the inflammatory process underlying Crohn’s disease [3,34]. The proinflammatory effects of TNF can increase the production of two additional key proinflammatory cytokines, IL-1β and IL-6 [3], which contribute to intestinal mucosal inflammation [35]. TFD and FFD showed significant inhibition of the production of IL-6 and IL-1β in DSS-induced IBD mice (Figure 2). Moreover, intestinal mucosal inflammation was reduced after TFD or FFD treatment in either DSS- or TNBS- induced IBD models (Figure 3). NF-κB has been identified as the master regulator of the immune-inflammatory response through regulating the transcription of genes that control inflammation [28]. In IBD patients, the activation of NF-κB promotes the production and secretion of proinflammatory cytokines TNFɑ, IL-6, and IL-1β [30,31]. The observation of reduced serum levels of IL-6 and IL-1β in the IBD model prompted us to examine whether NF-κB signaling was inhibited in animals treated with our drug candidates. The levels of p-p65 and p-IκBα, two major mediators of NF-κB signaling, were both reduced in TFD- or FFD-treated IBD mice. Collectively, these results indicated that TFD and FFD are protective against DSS- or TNBS- induced IBD mice, at least partially through inhibition of TNFα-activated NF-κB signaling.

TFD and FFD are oral antihistamines that target to histamine receptor-1 [32]. Histamine is one of the fundamental mediators of allergic disease through binding with specific surface receptors on target cells [36]. Four types of histamine receptors (H1-H4) have been recognized pharmacologically; these receptors have roles in the effects of histamine-related allergy (H1), modulating neurotransmitters release (H3) [37,38], and in modulating inflammation in autoimmune disease (H2, H4) [37]. Notably, H4 receptors are the most expressed histamine receptors in the gut, and H4’s involvement in inflammation during IBD has been demonstrated [39,40]. Importantly, some reports have indicated that histamine drives inflammation in colitis through targeting the H4 receptor. In contrast, other studies have shown that histamine receptor 4 deficiency aggravated inflammation in TNBS-induced colitis model mice [39,40]. The controversial effect of H4 receptor in IBD questions the strategy of pharmacological H4R blockade as a potential therapeutic option for patients suffering from IBD. Since the mechanism of action of fexofenadine is to selectively antagonize H1 receptors on the surface of cells on multiple different organ systems, we sought to determine whether FFD’s protective effect against IBD depends on targeting H1R through comparison of drug effects in WT and H1R KO mice with IBD. H1R KO mice exhibited impaired, but not abolished, protection from IBD relative to their WT counterparts across clinical scoring measures (Figure 5a–c), colon length retention (Figure 5g–i), serum cytokine content (Figure 6a–d), and histopathological analysis (Figure 6e–h). These results suggested that the protectiveness of FFD on IBD may not be H1R-dependent.

Histamine is a well-known critical molecule in allergic disease. However, histamine receptors have also been discovered in the human gut [41], and histamine can increase the secretion of Na and Cl ions and cause ion transport across the epithelial tissue in the gut through interacting with the H1 receptor [42]. Recently, H1R signaling was implicated in MAPK signaling and cAMP accumulation, inducing the proinflammatory gene expression [43]. MAP kinase’s function both upstream and downstream of signaling by TNFα receptors [44]. During the IBD progression, FFD ameliorated symptom severity through inhibiting TNFɑ signaling in our study, and the H1R deficiency may affect the MAPK signaling pathway and partially abolish the therapeutic inhibition of FFD on TNFα signaling, which may further affect the production of IL-6 and IL-1β (Figure 6a–d). Moreover, activation of H1R was found to be important for Th1 response as H1R deficiency engenders an exacerbated Th2 profile due to a diminished Th1 response [45]. IL-6 is a cytokine produced by several cell types, controlling Th1/Th2 differentiation, promoting Th2 differentiation, and inhibiting Th1 polarization [46]. H1R deficiency may abolish the inhibitory effect of FFD on IL-6 secretion, and subsequently, elevated IL-6 production may influence the Th1/Th2 response. As a proinflammatory cytokine [47], the higher level of IL-6 in H1R deficient mice may also induce the infiltration of inflammatory cells into the intestine (Figure 6e–h).

cPLA2 was isolated as a novel target of FFD in our recent study [15]. cPLA2 is comprised of six intracellular enzymes referring to as cytosolic PLA2 (cPLA2) α, -β, -γ, -δ, -ε, and -ζ, among them, cPLA2α is the most well-known group [48]. It is known that cPLA2 plays an important role in regulating autoimmune disease, neurodegenerative disease, and cancer [48,49,50,51,52] and has been implicated in inflammatory bowel disease [53,54]. Both cytosolic phospholipase A2 (cPLA2) and secreted phospholipase A2 (sPLA2) were reported to play a protective role from colitis mobilizing different types of arachidonic acid metabolites [53]. However, some studies report that cPLA2α was upregulated and activated in the development of colitis in the DSS-induced mouse model and preventing cPLA2 upregulation by i.v. administration of antisense oligonucleotides against cPLA2 prevented NF-κB activation and expression of proinflammatory proteins [54]. In our current DSS-induced mice model, cPLA2 deficient mice showed severe symptoms during the IBD development. No differences between DSS-induced cPLA2 KO mice and their WT controls were observed across body weight loss or stool score measures, while rectal bleeding was diminished in cPLA2 deficient mice (Appendix A). Our previous study showed that FFD could block TNF-stimulated cPLA2 activity and arachidonic acid production in the inflammatory arthritis mouse model [15]. Accordingly, we examined if the FFD-mediated anti-TNFɑ activity in IBD was cPLA2-dependent. Importantly, under cPLA2 deficiency, the therapeutic effect of FFD on IBD was practically abolished (Figure 7 and Figure 8). Although the deficiency of cPLA2 did not show significant effects on IBD development, the loss of FFD’s efficacy in DSS-induced cPLA2 mice indicated the importance of cPLA2 as a target of FFD-mediated stimulation of the TNF/TNFR- NF-κB anti-inflammatory pathway in the course of IBD.

Summarily, in our current study, we found that FFD, an over-the-counter drug, has a protective effect in IBD symptom progression through inhibiting TNFα signaling. This effect partially depends on H1R and predominately depends on cPLA2. The phospholipase A2 family is complex, with each catalytic subgroup suggested to hold unique regulatory and functional roles, and little is known about the in vivo functions of each subgroup excepting cPLA2ɑ [45]. Our current study suggests unrecognized functionality of FFD as a protective agent against IBD through a novel mechanism of targeting cPLA2, which may provide innovative interventions to develop cPLA2 targeting treatment in IBD and other inflammatory autoimmune diseases as well.

## 4. Materials and Methods

### 4.1. Mice

All animal studies were performed following institutional guidelines and approved by the Institutional Animal Care and Use Committee of New York University. Animals were housed on a 12-h light-dark cycle with ad libitum access to food and water in a specific pathogen-free environment. All animals were sex- and age-matched for experiments, typically at 8 weeks of age. H1R KO mice with C57BL/6J background were obtained from The Jackson Laboratory (Bar Harbor, ME, USA). C57BL/6J wild-type mice were used as a wild-type control for H1R KO mice. cPLA2 KO mice originated from Joseph V Bonventre at Brigham and Women’s Hospital and were provided, with Dr. Bonventre’s permission, by Dr. Naikui Liu’s lab at Indiana University School of Medicine. These cpla2 KO mice are on a C57BL/6 and 129 mixed background and were obtained through heterozygous crosses. Wild-type littermates were used as controls for cPLA2 KO mice.

### 4.2. DSS-Induced Colitis Model

Dextran Sulfate Sodium (DSS)-induced colitis model was established in 8-week-old mice via administration of drinking water containing 3% DSS for 5 days and followed by normal drinking water for 3 days. Fexofenadine (0.4, 2, 10 mg/kg), Terfenadine (2, 10, 50 mg/kg), or 5-ASA (50 mg/kg) were administered orally 3 days before model induction and continued daily for 10 days after which mice were sacrificed. TFD and FFD diluent (water/ethanol/2% acetic acid 8:3:1 (*v/v*)) was used as control. Based on a previously published scoring system, bodyweight, stool consistency, and rectal bleeding were recorded daily [46]. Scoring criteria, which summarized in the Appendix A, are as follows: Weight loss: 0 = less than 1%, 1 = between 5% and 10%, 2 = between 10% and 15%, 3 = between 15% and 20%, 4 = over 20%. Stool consistency: 0 = normal, 2 = loose stool, 4 = diarrhea. Rectal bleeding: 0 = negative, 2 = blood trace, 4 = gross blood. Colon length was measured, and colons and the sera were processed for further analysis.

### 4.3. TNBS Induced Colitis Model

Before establishing the trinitro-benzene-sulfonic acid (TNBS) induced colitis model, 8-week-old mice were dermally pre-sensitized using 150 μL 1% TNBS C57BL/6. A week later, 100 μL 2.5% TNBS was delivered rectally. Oral delivery of Fexofenadine (0.4, 2, 10 mg/kg), Terfenadine (2, 10, 50 mg/kg), 5-aminosalicylic acid (5-ASA, 50 mg/kg, serving as a positive control) were initiated three days before model initiation and continued daily for 5 days. Body weights were recorded daily. After sacrifice, colon length was measured, and colons and the sera were processed for further analysis. Six mice were used for each group.

### 4.4. Histological Analysis 

Mouse intestine tissues were fixed in 4% formaldehyde and embedded in paraffin. Serial 6µm sections were performed Hematoxylin and Eosin (H&E) staining to score the inflammation. For analysis of colitis, tissues were blindly scored by a trained observer, and the scores were the combination of inflammatory cell infiltration and intestinal wall structure integrity. The higher scores indicated more serious inflammation. Scores of inflammatory cell infiltration were: 0 = normal, 1 = inflammatory cell only infiltrated to the mucosa, 2 = inflammatory cell reached to mucosa and sub-mucosa, 3 = inflammatory cell was found in the whole intestinal wall. Intestinal wall structure integrity scores were: 0 = normal, 1 = inflammatory cell was locally infiltrated, 2 = focally formed ulceration, 3 = extensively formed ulceration with or without granulation tissue or pseudo-polyps. Images were obtained by digital microscope (Axio Scope A.1, Carl Zeiss, LLC, Jena, Germany).

### 4.5. Immunohistochemistry Staining 

Mouse intestine tissues were fixed in 4% paraformaldehyde and embedded in paraffin. Tissue sections were then prepared and performed with immunohistochemistry staining with phospho-p65 (p-p65) (Cell Signaling Technology, Danvers, MA, USA, 93H1, 3033S) and phospho-IκB-ɑ (p-IκB-ɑ) (Cell Signaling Technology, Danvers, MA, USA, 14D4, 2859S), to analysis the phosphorylation state of phospho-p65 and -IκB-ɑ. The positive cell number was counted manually. The quantification was done by the mean value of positive cell number ratio between the treatment group and control group.

### 4.6. ELISA 

Cytokine levels of IL-1β (88-7013, Invitrogen, Waltham, MA, USA) and IL-6 (88706476, Invitrogen, Waltham, MA, USA) in sera from mouse models were detected by sandwich ELISA according to product specifications. Sera were separated from whole blood by centrifuge freshly collected blood from mice at 3000 rmp for 10 min.

### 4.7. Statistical Analysis 

For comparison of treatment groups, we performed a t-test or one-way ANOVA (where appropriate). All statistical analysis was performed using GraphPad Prism 7 Software. Data are shown as mean ± SE, * *p* < 0.05, ** *p* < 0.01, *** *p* < 0.001.

## 5. Conclusions

In this study, we first demonstrated the therapeutic effects of FFD against IBD using both dextran sulfate sodium- and 2, 4, 6-trinitrobenzene sulfonic acid-induced IBD models. By leveraging the knockout mice for cPLA2 and histamine H1 receptor, a well-known long target of Fexofenadine, we demonstrated that Fexofenadine completely lost its therapeutic effects in cPLA2 deficient mice, but not in histamine H1 receptor-deficient mice. Our findings demonstrate the therapeutic effects of over-the-counter drug Fexofenadine in treating IBD and provide first in vivo evidence showing that cPLA2 is required for Fexofenadine’s therapeutic effects in IBD and probably other inflammatory and autoimmune diseases as well. In brief, this study broadens the clinical application of Fexofenadine and provides new insights into the understanding of Fexofenadine’s action and the underlying mechanism.

## Figures and Tables

**Figure 1 ijms-22-11155-f001:**
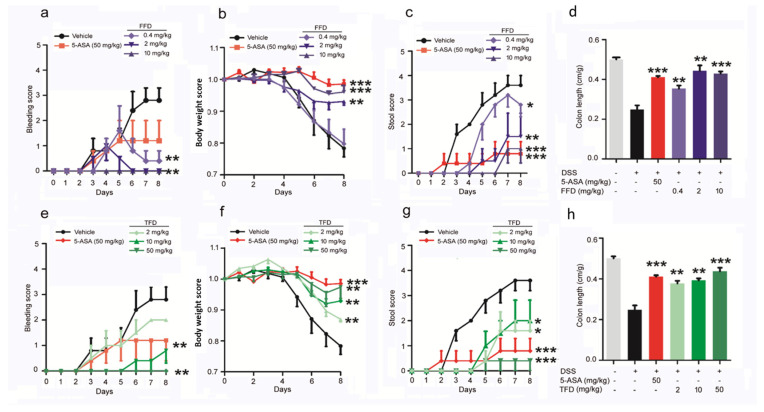
Clinical scores of Fexofenadine (FFD)-treated or Terfenadine (TFD)-treated, or controls in DSS model mice. DSS-induced colitis model was established in 8-week-old C57BL/6 mice with ad libitum access to drinking water containing 3% DSS for 10 days, followed by normal drinking water for 3 days. 5-ASA (serving as a positive control, 50 mg/kg), Fexofenadine (FFD, 0.4, 2, 10 mg/kg), and Terfenadine (TFD, 2, 10, 50 mg/kg) were orally delivered beginning 3 days before delivery of DSS in drinking water and continuing until the mice were sacrificed. (**a**–**d**) Clinical score of controls and FFD-treated groups. (**a**) Bleeding scores of controls and FFD-treated groups. (**b**) Body weights of controls and FFD-treated groups. (**c**) Stool score of controls and FFD-treated groups. (**d**) Colon lengths of controls and FFD-treated groups. (**e**–**h**) Clinic score of controls and TFD-treated groups. (**e**) Bleeding scores of controls and TFD-treated groups. (**f**) Body weights of controls and TFD-treated groups. (**g**) Stool score of controls and TFD-treated groups. (**h**) Colon lengths of controls and TFD-treated groups. *n* = 6 per group. Statistical analysis was done to compare the treatment groups and vehicle groups. Data are mean ± SE; * *p* < 0.05, ** *p* < 0.01, *** *p* < 0.001.

**Figure 2 ijms-22-11155-f002:**
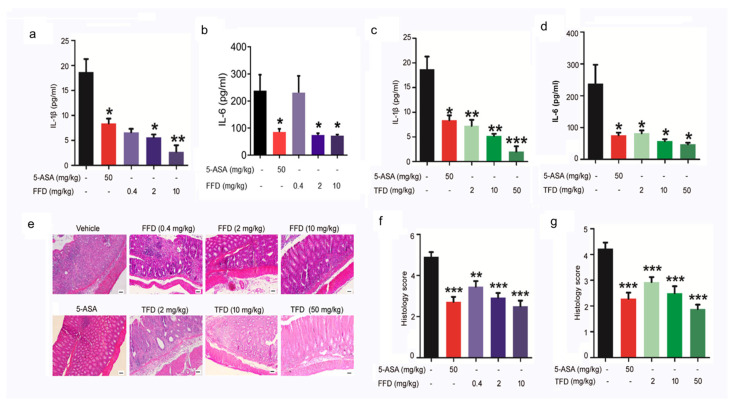
Inflammatory response of Fexofenadine (FFD)-treated or Terfenadine (TFD)-treated, or controls in DSS model mice. a-d. Serum levels of IL-1β and IL-6. (**a**) Serum level of IL-1β in ctrl, or 5-ASA or FFD-treated group. (**b**) Serum level of IL-6 in ctrl, or 5-ASA or FFD-treated group. (**c**) Serum level of IL-1β in ctrl, or 5-ASA or TFD-treated group. (**d**) Serum level of IL-6 in ctrl, or 5-ASA or TFD-treated group. (**e**–**g**) H&E staining and its quantitative analysis in colon tissue sections in ctrl or 5-ASA, or FFD, or TFD-treated group. Statistical analysis was done to compare the treatment groups and vehicle groups. *n* = 6 per group. Data are mean ± SE; * *p* < 0.05, ** *p* < 0.01, *** *p* < 0.001. (Scale bar, 100 μm).

**Figure 3 ijms-22-11155-f003:**
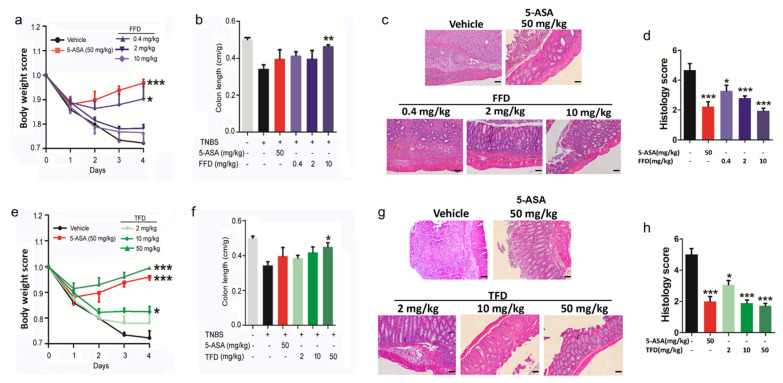
Effects of Fexofenadine (FFD) or Terfenadine (TFD) on TNBS-induced acute colitis model. Beginning 3 days before intrarectal TNBS injection, drugs were orally delivered until sacrifice at 5 days. 2.5% TNBS was administered by intrarectal injection to 8-week-old C57BL/6 mice. (**a**) Body weights of controls and FFD-treated groups. (**b**) Colon lengths of controls and FFD-treated groups. (**c**) H&E staining of colons from the FFD-treated group. (**d**) The quantification of the histological score. (**e**) Body weights of controls and TFD-treated groups. (**f**) Colon lengths of controls and TFD-treated groups. (**g**) H&E staining of colons from the TFD-treated group. (**h**) The quantification of the histological score. Statistical analysis was done to compare the treatment groups and vehicle groups. *n* = 6 per group. Data are mean ± SE; * *p* < 0.05, ** *p* < 0.01, *** *p* < 0.001. (Scale bar, 100 μm).

**Figure 4 ijms-22-11155-f004:**
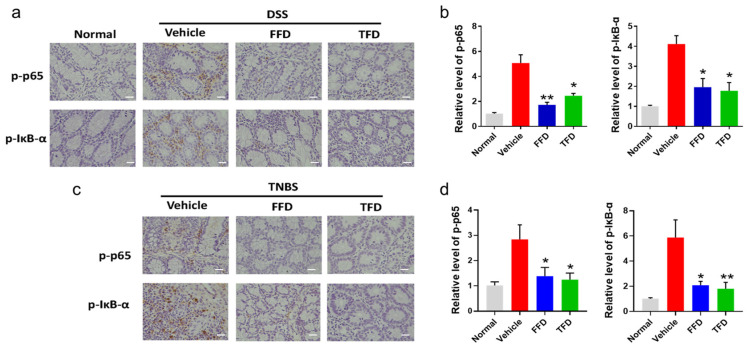
Immunohistochemistry (IHC) analysis of p-p65 and p-IκB-α in colon tissues. (**a**) IHC staining of p-p65 and p-IκB-α in paraffin-embedded samples collected from DSS model mice. (**b**) Statistical analysis of a. (**c**) IHC staining of p-p65 and p-IκB-α in paraffin-embedded samples collected from TNBS model mice. (**d**) Statistical analysis of c. Terfenadine and Fexofenadine are indicated with TFD (10 mg/kg) and FFD (2 mg/kg), respectively. Statistical analysis was done to compare the treatment groups and vehicle groups. *n* = 6 per group. Data are mean ± SE; * *p* < 0.05, ** *p* < 0.01. (Scale bar, 50 μm).

**Figure 5 ijms-22-11155-f005:**
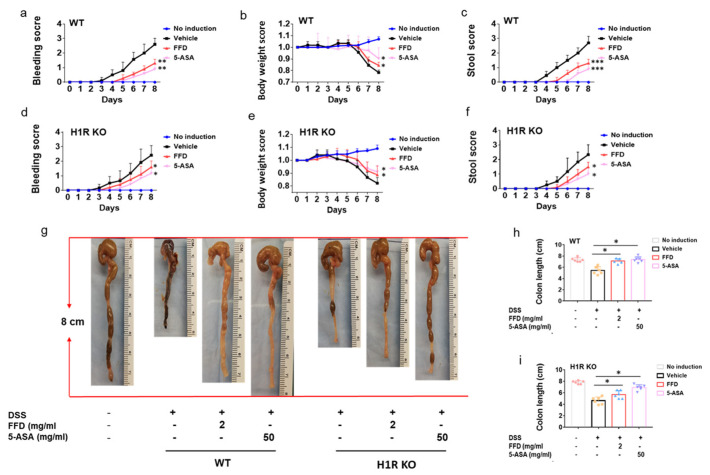
Clinical score of Fexofenadine (FFD)-treated and control groups in DSS-induced H1R knockout mice. (**a**–**c**) Clinical score of controls and FFD-treated WT groups. (**a**) Bleeding scores of controls and FFD-treated WT groups. (**c**) Body weights of controls and FFD-treated WT groups. (**d**) Stool score of controls and FFD-treated WT groups. (**d**–**f**) Clinical score of controls and FFD-treated H1R KO groups. (**d**) Bleeding scores of controls and FFD-treated H1R KO groups. (**e**) Body weights of controls and FFD-treated H1R KO groups. (**f**) Stool score of controls and FFD-treated H1R KO groups. (**g**–**i**) Colon lengths of controls and FFD-treated WT or H1R KO groups and the quantification of colon lengths. Statistical analysis was done to compare the treatment groups and vehicle groups. *n* = 6 per group. Data are mean ± SE; * *p* < 0.05, ** *p* < 0.01, *** *p* < 0.001.

**Figure 6 ijms-22-11155-f006:**
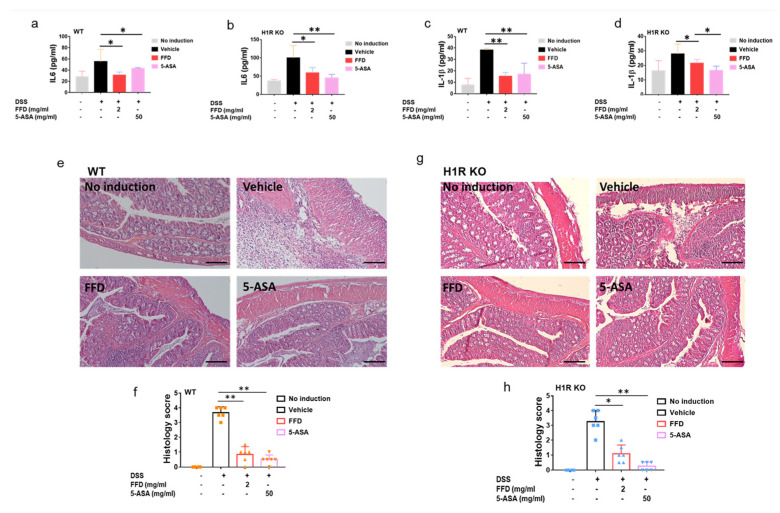
Inflammatory response of FFD-treated or controls in DSS-induced WT and H1R KO mice. (**a**–**d**) Serum levels of IL-1β and IL-6. (**a**) Serum level of IL-6 in ctrl, or 5-ASA or FFD-treated WT group. (**b**) Serum level of IL-6 in ctrl, or 5-ASA or FFD-treated H1R KO group. (**c**) Serum level of IL-1β in ctrl, or 5-ASA or FFD-treated WT group. (**d**) Serum level of IL-1β in ctrl, or 5-ASA or FFD-treated H1R KO group. (**e**,**f**) H&E staining and its quantitative analysis in colon tissue sections in ctrl or 5-ASA, or FFD-treated WT group. (**g**,**h**) H&E staining and its quantitative analysis in colon tissue sections in ctrl or 5-ASA, or FFD-treated H1R KO group. Statistical analysis was done to compare the treatment groups and vehicle groups. *n* = 6 per group. Data are mean ± SE; * *p* < 0.05, ** *p* < 0.01. (Scale bar, 100 μm).

**Figure 7 ijms-22-11155-f007:**
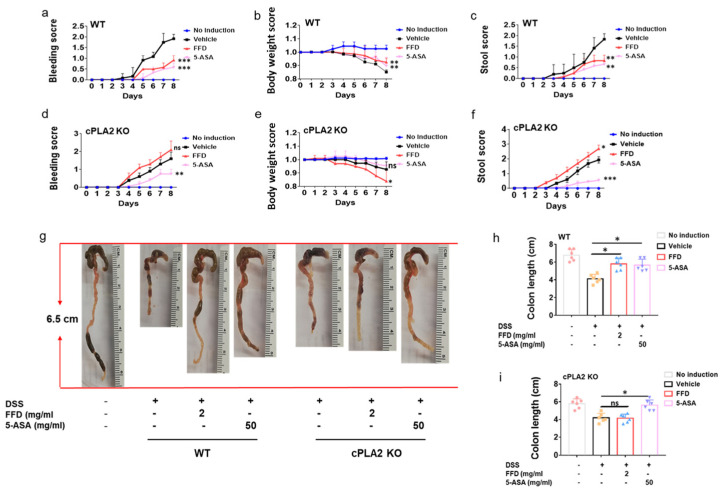
Clinical score of Fexofenadine (FFD)-treated and control groups in DSS-induced cPLA2 KO mice. (**a**–**c**) Clinical score of controls and FFD-treated WT groups. (**a**) Bleeding scores of controls and FFD-treated WT groups. (**c**) Body weights of controls and FFD-treated WT groups. (**d**) Stool score of controls and FFD-treated WT groups. (**d**–**f**) Clinical score of controls and FFD-treated cPLA2 KO groups. (**d**) Bleeding scores of controls and FFD-treated cPLA2 KO groups. (**e**) Body weights of controls and FFD-treated cPLA2 KO groups. (**f**) Stool score of controls and FFD-treated cPLA2 KO groups. (**g**–**i**) Colon lengths of controls and FFD-treated WT or cPLA2 KO groups and the quantification of colon lengths. Statistical analysis was done to compare the treatment groups and vehicle groups. *n* = 6 per group. Data are mean ± SE; * *p* < 0.05, ** *p* < 0.01, *** *p* < 0.001, ns: not significant.

**Figure 8 ijms-22-11155-f008:**
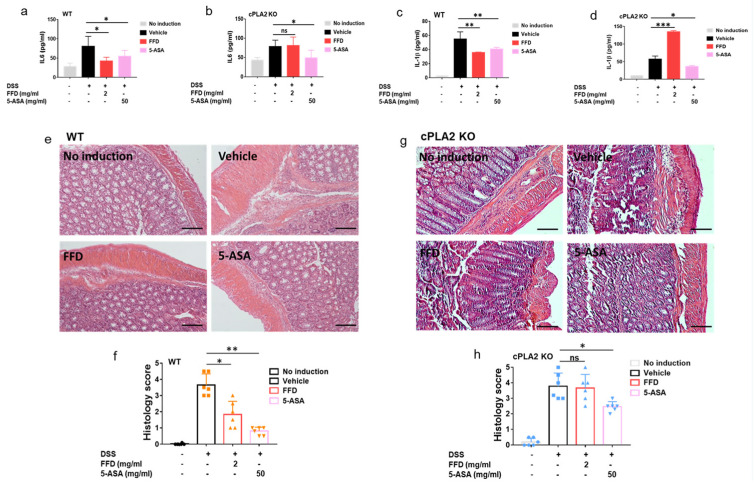
Inflammatory response of Fexofenadine (FFD)-treated or controls in DSS-induced WT or cPLA2 KO mice. (**a**–**d**) Serum levels of IL-1β and IL-6. (**a**) Serum level of IL-6 in ctrl, or 5-ASA or FFD-treated WT group. (**b**) Serum level of IL-6 in ctrl, or 5-ASA or FFD-treated cPLA2 KO group. (**c**) Serum level of IL-1β in ctrl, or 5-ASA or FFD-treated WT group. (**d**) Serum level of IL-1β in ctrl, or 5-ASA or FFD-treated cPLA2 KO group. (**e**,**f**) H&E staining and its quantitative analysis in colon tissue sections in ctrl or 5-ASA, or FFD-treated WT group. (**g**,**h**) H&E staining and its quantitative analysis in colon tissue sections in ctrl or 5-ASA, or FFD-treated cPLA2 KO group. Statistical analysis was done to compare the treatment groups and vehicle groups. *n* = 6 per group. Data are mean ± SE; * *p* < 0.05, ** *p* < 0.01, *** *p* < 0.001. (Scale bar, 100 μm). ns: not significant.

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
