# Peer review of "Cytosolic Phospholipase A2 Is Required for Fexofenadine’s Therapeutic Effects against Inflammatory Bowel Disease in Mice"

_ijms, 2021, doi:10.3390/ijms222011155_

Round 1

Reviewer 1 Report

 Remarks to the Author:

Zhao et al. investigate the therapeutic efficacy of Fexofenadine (FFD) in murine colitis models in their manuscript. The findings revealed that the protective effect of FFD against colitis was dependent on cytosolic phospholipase A2 (cPLA2). H1R-deficient mice and cPLA2 mice were used in this study to reveal that Fex targets cPLA2 to inhibit intestinal inflammation in DSS-induced colitis. However, the cellular mechanism is not novel since this group published a well-performed study (PMID: 31302596) in 2019 revealing cPLA2 as a novel fexofenadine target. However, this work has clinical implications, since it reveals the mechanism through which FFD inhibits intestinal inflammation in vivo, which depends on cPLA2. As stated below, there are several concerns with the data given.

Major points,

  1. Inflammatory bowel disease (IBD) refers solely to chronic intestinal disease (mostly UC and CD) in both animals and humans, and does not include any chemically induced acute murine colitis models. Please correct your statement that IBD refers to murine colitis models induced by DSS and TNBS (For example, Line 17, Line 69, and so on.)
  2. Fig1-7, Data are given as mean+/- SD; however, the reviewer believes these data sets represent the mean of individual mice; hence, data should be reported as mean+/-SE. Please seek the advice of a statistician to ensure that the appropriate statistical analysis is utilized in this paper.
  3. Because the resolution of the histological picture shown in the figure is limited, and the main focus of this work is on understanding the therapeutic impact of FFD on intestinal inflammation, the authors should consult a pathologist to confirm your histological interpretation.
  4. No information was provided on the number of repeats of the experiment (Fig 1-Fig. 8)
  5. It would be great if readers could interpret from the results in Fig. 7 that the DSS-induced colitis model was performed on WT and cPLA2 KO animals in the same experiment.
  6. Because the data display in Fig 7 is incomplete, the reviewer advise that the authors directly plot the WT and cPLA2 KO together to see if there is a difference between these two groups. Whether intrinsic cPLA2 deficiency was necessary for intestinal inflammation in the absence of FFD and 5-ASA. If not, could you please explain why cPLA2 is not required in maintaining intestinal inflammation whereas FFD can exert immunomodulatory effects on intestinal inflammation via the cPLA2 pathway?

Author Response

Dear reviewer,

Thank you very much for your enthusiasm for this study, and particularly for your insightful and helpful comments provided toward improvement of the paper. We believe that the revised manuscript has benefited as a result. We have prepared a point-by-point reply to address your concerns. In our point-by-point response, reviewers’ comments are in italics, while our responses are in bold. The attachment is the highlighted version.

Reviewer 

Zhao et al. investigate the therapeutic efficacy of Fexofenadine (FFD) in murine colitis models in their manuscript. The findings revealed that the protective effect of FFD against colitis was dependent on cytosolic phospholipase A2 (cPLA2). H1R-deficient mice and cPLA2 mice were used in this study to reveal that Fex targets cPLA2 to inhibit intestinal inflammation in DSS-induced colitis. However, the cellular mechanism is not novel since this group published a well-performed study (PMID: 31302596) in 2019 revealing cPLA2 as a novel fexofenadine target. However, this work has clinical implications, since it reveals the mechanism through which FFD inhibits intestinal inflammation in vivo, which depends on cPLA2. As stated below, there are several concerns with the data given.

Response: We thank you for the positive comments and particularly for your excellent points below that further strengthen the paper.

Major points,

  1. Inflammatory bowel disease (IBD) refers solely to chronic intestinal disease (mostly UC and CD) in both animals and humans, and does not include any chemically induced acute murine colitis models. Please correct your statement that IBD refers to murine colitis models induced by DSS and TNBS (For example, Line 17, Line 69, and so on.)

Response: As suggested, we corrected our statement (Please see the highlighted text in page 2, line 23, 25, 32; page 4, line 70, 75; page 9, line 176, 187, and so on)

  1. Fig1-7, Data are given as mean+/- SD; however, the reviewer believes these data sets represent the mean of individual mice; hence, data should be reported as mean+/-SE. Please seek the advice of a statistician to ensure that the appropriate statistical analysis is utilized in this paper.

Response: Thanks for the excellent point! The data sets in this paper represent the mean of individual mice, we corrected ‘mean+/- SD’ to ‘mean+/- SE’ (Please see the highlighted text in each figure legend).

  1. Because the resolution of the histological picture shown in the figure is limited, and the main focus of this work is on understanding the therapeutic impact of FFD on intestinal inflammation, the authors should consult a pathologist to confirm your histological interpretation.

Response: Thanks for the suggestion! With the help of a pathologist, we have added more detail concerning historical analysis (Please see the highlighted text in page 6, line 117)

  1. No information was provided on the number of repeats of the experiment (Fig 1-Fig. 8)

Response: The number of mice, representing biological replicates, is now included in each figure legend.

  1. It would be great if readers could interpret from the results in Fig. 7 that the DSS-induced colitis model was performed on WT and cPLA2 KO animals in the same experiment.

Response: Thanks for the comment! C57BL/6J mice were used as a control for H1R KO mice.  Our cpla2 strain is on a mixed C57BL/6x129 background; WT controls for cpla2 KO mice were obtained as littermates from heterozygous crosses. Since different WTs were used as the controls of H1R KO and cPLA2 KO mice, we have separated this data out. Fig.7 does show results from the DSS-induced colitis model as performed in cpla2 KO mice and their WT littermates. In Fig. 7, we have kept the WT and cpla2 KO graphs separate to allow for legibility.

  1. Because the data display in Fig 7 is incomplete, the reviewer advise that the authors directly plot the WT and cPLA2 KO together to see if there is a difference between these two groups. Whether intrinsic cPLA2 deficiency was necessary for intestinal inflammation in the absence of FFD and 5-ASA. If not, could you please explain why cPLA2 is not required in maintaining intestinal inflammation whereas FFD can exert immunomodulatory effects on intestinal inflammation via the cPLA2 pathway

Response: We greatly appreciate reviewer for these insightful points! As suggested, we plot the WT and cPLA2 KO together to compare the difference between these two groups. As showed in the supplementary Table 2, the deficiency of cPLA2 showed only a protective effect on the rectal bleeding in the DSS-induced IBD mice, however, the therapeutic effect of FFD on IBD was practically lost in cPLA2 KO mice (Please see the supplementary Table 2 and the highlighted text in page 16, line 335-346).

Reviewer 2 Report

In this manuscript, Xiangli Zhao and colleagues assessed the therapeutic effect of fexofenadine in colitis models and decipher the mechanisms involving H1R and cPLA2.

While most of the conclusions are supported by the data presented, there is however some overstatements that need to be carefully edited (see below). I also bring some comments to the authors attention.

Major comment: The author should comment on the role of Histamine receptor 4 which is the most expressed histamine receptor in the gut immune cells and epithelial cells and also because its involvement during IBD has been demonstrated. To conclude on the role of histamine receptor, author should investigate the role of the 3 other histamine receptors expressed in gut.

Minor points:

Reference number 19, cited in the text as a reference for FFD in IBD is wrong.

Line 173: please modify, “the bleeding and stool scores were significantly higher (not lower) in vehicle group as compared with the no induction group…”

Line 194: “FFD-treatment was even more effective than 5ASA treatment”, please moderate this comment. IL1b and IL6 secretion is not really different between FFD and 5ASA treatments in WT mice.  

Line 198: please mention that you are speaking about the histological observation/score.

Lines 198, 199: the aim of this part is to assess the efficacy of FFD in colitis in WT vs H1R deficient mice, not to compare FFD and 5ASA.

Line 200: please moderate your comment, figure 6 g-h does not show statistically difference between FFD and 5ASA.

Line 219: please detail the results of FFD treatment in cPLA2 KO mice.

Line 227: please moderate your comment. The impact of 5ASA on histology score is not abrogated in cPLA2 KO mice compared to WT mice, but just less pronounced.

Line 279: the difference in colitis development between WT and H1R KO mice is difficult to appreciate based on the figures. Were the experiments performed together? Is there any statistical analysis performed between WT and H1R KO mice?

Figure 5g: there is no picture from H1R non treated mice.

Line 282-283: I disagree, there is no difference in IL1b and IL6 secretion between FFD and 5ASA in WT mice.

Line 284: There is no data that allows you to assert that FFD effect is partly due to H1R. In each tested parameter, FFD treatment is always significant compared to vehicle in KO mice.

Line 286: H4 receptor is also expressed in the gut.  

Materials and methods section: the “mice” paragraph is a little bit confusing; were the WT animals, used as control for the cPLA2 KO mice experiments, on a mixed C57B/6 and 129 background as the cPLA2 mice?

Figures: Please explain the calculation on the body weight graph, why the starting point is at 1, while the scoring criteria start at 0 for no weight loss? Maybe state “body weight loss score” instead of “body weight” in the figures.

Statistical analysis: please state in each figure legend the groups which were compared for statistical analysis. I assume for example in figure 1a that the 2 stars in FFD groups is a comparison with the vehicle group?

Author Response

Dear reviewer,

Thank you very much for your enthusiasm for this study, and particularly for your insightful and helpful comments provided toward improvement of the paper. We believe that the revised manuscript has benefited as a result. We have prepared a point-by-point reply to address your concerns. In our point-by-point response, reviewers’ comments are in italics, while our responses are in bold. The attachment is the highlighted version.

Reviewer :

In this manuscript, Xiangli Zhao and colleagues assessed the therapeutic effect of fexofenadine in colitis models and decipher the mechanisms involving H1R and cPLA2.

While most of the conclusions are supported by the data presented, there is however some overstatements that need to be carefully edited (see below). I also bring some comments to the authors attention.

 Response: We appreciate you for the positivity and recommendation.

Major comment: The author should comment on the role of Histamine receptor 4 which is the most expressed histamine receptor in the gut immune cells and epithelial cells and also because its involvement during IBD has been demonstrated. To conclude on the role of histamine receptor, author should investigate the role of the 3 other histamine receptors expressed in gut.

Response: We thank you for the wonderful suggestions! As suggested, we discussed the role of histamine receptor family, including H1, H2, H3, and H4, in the discussion part (Please see the highlighted text in page 14, line 293-302).

Minor points:

Reference number 19, cited in the text as a reference for FFD in IBD is wrong.

Response: We apologize for the error. We have checked the text and references carefully and corrected the reference 19 (Please see new the highlighted text in page 20, reference 19).

Line 173: please modify, “the bleeding and stool scores were significantly higher (not lower) in vehicle group as compared with the no induction group…”

Response: As suggested, we have modified the sentence (Please see the highlighted text in page 10, line 217-218).

Line 194: “FFD-treatment was even more effective than 5ASA treatment”, please moderate this comment. IL1b and IL6 secretion is not really different between FFD and 5ASA treatments in WT mice. 

Response: Done as suggested (Please see the highlighted text in page 11, line 230-231).

Line 198: please mention that you are speaking about the histological observation/score.

Response: Done as suggested (Please see new the highlighted text in page 11, line 234-235).

Lines 198, 199: the aim of this part is to assess the efficacy of FFD in colitis in WT vs H1R deficient mice, not to compare FFD and 5ASA.

Response: Thanks for the point, and we have edited this in the text (Please see the text in page 11, line 229-235).

Line 200: please moderate your comment, figure 6 g-h does not show statistically difference between FFD and 5ASA.

Response: As suggested, we have moderated our comment (Please see the text in page 11, line 234-235).

Line 219: please detail the results of FFD treatment in cPLA2 KO mice.

Response: As suggested, we have detailed the results of FFD treatment in cPLA2 KO mice (Please see the text in page 12, line 246-247, line 252-255).

Line 227: please moderate your comment. The impact of 5ASA on histology score is not abrogated in cPLA2 KO mice compared to WT mice, but just less pronounced.

Response: Done as suggested (Please see the text in page 12, line 250-251).

Line 279: the difference in colitis development between WT and H1R KO mice is difficult to appreciate based on the figures. Were the experiments performed together? Is there any statistical analysis performed between WT and H1R KO mice?

Response: We appreciate reviewer for these great points! As suggested, we combined the WT and H1R KO together to compare the difference between these two groups. As showed in the supplementary Table 2, no difference was observed between DSS-induced WT and H1R KO mice (Please see the supplementary Table 2 and the highlighted text in page 11, line 223-226).

Figure 5g: there is no picture from H1R non treated mice.

Response: H1R non treated mice, the No induction group, is indicated in Figure 5.

Line 282-283: I disagree, there is no difference in IL1b and IL6 secretion between FFD and 5ASA in WT mice.

Response: Thanks and we have edited this comment (Please see the highlighted text in page 13, line 279-282).

Line 284: There is no data that allows you to assert that FFD effect is partly due to H1R. In each tested parameter, FFD treatment is always significant compared to vehicle in KO mice.

Response: Thanks for the comments, and we have moderated the statement concerned (Please see the highlighted text in page 14, line 308-309).

Line 286: H4 receptor is also expressed in the gut. 

Response: Thanks for the comments, and we discussed the role of histamine receptors 4 in the gut of IBD model (Please see the highlighted text in page 14, line 293-302).

Materials and methods section: the “mice” paragraph is a little bit confusing; were the WT animals, used as control for the cPLA2 KO mice experiments, on a mixed C57B/6 and 129 background as the cPLA2 mice?

Response: Thanks for the great point! We have detailed this part. The WT mice with C57BL/6J background were used as the control for H1R KO mice, and WT mice with mixed C57B/6 and 129 background were used as the control of cPLA2 KO mice.

Figures: Please explain the calculation on the body weight graph, why the starting point is at 1, while the scoring criteria start at 0 for no weight loss? Maybe state “body weight loss score” instead of “body weight” in the figures.

Response: As suggested, we changed the “body weight” to the “body weight loss score” (Please see the Figure 1, 3, 5 and 7).

Statistical analysis: please state in each figure legend the groups which were compared for statistical analysis. I assume for example in figure 1a that the 2 stars in FFD groups is a comparison with the vehicle group?

Response: Thanks and we have stated it in each figure legend.
